# Increases in income-support payments reduce the demand for charity: A difference-in-difference analysis of charitable-assistance data from Australia over the COVID-19 pandemic

Christine Ablaza[1,2]*, Francisco Perales[1,2], Cameron Parsell[1,2], Nathan Middlebrook[3], Richard N. S. Robinson[4], Ella Kuskoff[1,2], Stefanie Plage[1,2]

1 School of Social Science, The University of Queensland, St Lucia, QLD, Australia, 2 ARC Centre of Excellence for Children and Families Over the Life Course, The University of Queensland, St Lucia, QLD, Australia, 3 St. Vincent de Paul Society Queensland, South Brisbane, QLD, Australia, 4 UQ Business School, The University of Queensland, St Lucia, QLD, Australia

* c.ablaza@uq.edu.au

## Abstract

Charities play an increasingly important role in helping people experiencing poverty. However, institutionalized charity shifts the burden of poverty reduction away from the state and exposes recipients to stress and stigma. In this paper, we examine whether the need for institutionalized charity can be offset through enhanced state support. As in other countries, the Australian government responded to the COVID-19 pandemic by substantially increasing the level of income support to citizens through several temporary payments. We draw on this natural experiment and time-series data from the two largest charity organizations in Queensland, Australia to examine how these payments altered the demand for institutionalized charity. We model these data using difference-in-difference regression models to approximate causal effects. By exploiting the timing and varying amounts of the payments, our analyses yield evidence that more generous income support reduces reliance on charity. Halving the demand for charity requires raising pre-pandemic income-support by AUD $42/day, with supplements of approximately AUD$18/day yielding the greatest return on investment.

## Introduction

Two decades into the 21$^{st}$ century, charity remains an ever-present and inescapable reality in the lives of many. Indeed, a growing number of citizens in advanced welfare states resort to charity organizations, including food banks, to meet their basic needs. This increase has been observed across countries with different cultural values and state-support practices, such as the US [1], UK [2], Australia [3], and Finland [4]. Charity and community-based ground-up models of assisting people in need are sometimes positioned as representing a natural form of help

**Data Availability Statement:** All relevant data are within the paper and its Supporting Information files.

**Funding:** This research was supported by (i) the Australian Research Council Centre of Excellence for Children and Families over the Life Course (project number CE200100025), (ii) an Australian Future Fellowship Research Grant FT180100250), https://www.arc.gov.au/ and (iii) the St. Vincent de Paul Society Queensland, https://qld.vinnies.org.au/. There was no additional external funding received for this study. The funders had no role in study design, data collection and analysis, decision to publish, or preparation of the manuscript.

**Competing interests:** We have read the journal's policy and the authors of this manuscript have the following competing interests to declare: N.M. works as Community Engagement Manager at St. Vincent de Paul Society Queensland (whose data was used in this study). The University of Queensland and St. Vincent de Paul Society Queensland have an ongoing research partnership, but the Society was not involved in the conceptualization, analysis, and preparation of this manuscript. The views expressed in this article do not necessarily represent the views of St. Vincent de Paul Society or The Salvation Army. This does not alter our adherence to PLOS ONE policies on sharing data and materials.

and a manifestation of a connected and caring community [5–7]. Others advocate for institutionalized charity—or charity provided by organizations rather than individuals—over state support out of concern that the latter disincentivizes paid employment [8, 9]. For people who live in poverty, however, accessing charity is less than ideal. Reliance on charity brings about feelings of stress, shame, and stigmatization, and those who ask for charity liken the experience to 'begging', especially when the charitable make morally based judgements on the recipients' deservingness [10, 11]. Further, institutionalized charity cannot fully address the extant needs of people in poverty [12], while also diverting attention from structural failure [13].

Multiple voices in the academic and service-delivery sectors strongly argue that, in advanced welfare states, the need for people to rely on charity and goodwill to eat, pay rent, or dress their children is the product of deliberate government decisions [14]. Some scholars maintain that, despite government rhetoric about financial constraints, the state has sufficient means to provide more generous income-support payments to people in need that would prevent this situation [15]. Beginning with Marshall [16], others emphasize governments' moral responsibility to appropriately support citizens, as well as individuals' basic social rights to an adequate level of income. In recent times, this principle is reflected in calls for a universal basic income that would negate the role of charity [17–19]. Recent changes in Australian welfare policy have reignited these debates about the role of the state in meeting citizen need. In response to the COVID-19 pandemic, the Australian government significantly increased the number, amount, and coverage of payments aimed at supporting the most vulnerable segments of society [20]. This response is consistent with policy strategies adopted internationally, where governments have engaged in deficit spending to support cash transfers aimed at mitigating the effects of unemployment—an approach referred to as 'emergency Keynesianism' [21, 22].

While these debates have been front and center in academic and policy circles, whether or not insufficient support to people in need drives individuals' necessity to resort to charity remains an open empirical question. In this study, we address this knowledge gap and provide a novel answer to this important question. Specifically, we leverage the unique natural experiment arising from these temporary changes in Australia's welfare policy to examine how increases in income support affect the demand for institutionalized charity. We focus specifically on emergency relief, a type of charitable assistance aimed at providing people in immediate need with the most basic necessities (e.g., food, clothing, personal-hygiene items, or rent money). Applying difference-in-difference and triple-difference estimation to daily-level data on charitable assistance, we exploit the timing of these reforms and shifting levels of income support to gather unique evidence of an inverse relationship between government income support and charity reliance.

## Charity, poverty, and state support in contemporary Australia

Australia constitutes a valuable case study to examine the interrelationships between income support and institutionalized charity. Deeming [23] describes Australia as a radical interventionist welfare state characterized by a heavy reliance on means-testing, strict welfare conditionalities, and a relatively high volume of transfers from high- to low-income groups. In 2017, the country's share of public social spending to GDP (including for social protection) was nevertheless significantly lower than the OECD average (17% compared to 20%) (*S1 Fig in S1 File*). This translates to low benefit levels for certain groups, such as the unemployed.

Indeed, Australia has the third lowest unemployment-benefit replacement rate in the OECD (*S2 Fig in S1 File*). Between 2017 and 2018, a person receiving Youth Allowance (the unemployment benefit for citizens aged 16 to 21) was entitled to AUD$289 per week—AUD

$168 per week below the poverty line. Similarly, JobSeeker (the unemployment benefit for citizens aged 22 to 65) paid eligible recipients AUD$340 per week—AUD$117 per week below the poverty line [24]. As a result, poverty rates in Australia have remained stagnant at 13% since 2000, while the depth of poverty has progressively increased [24].

Importantly, the low benefit levels characterizing Australia's welfare system are not driven by a financial imperative for austerity. Prior to 2020, Australia was the only high-income country to have enjoyed 28 continuous years of GDP growth. In 2020, it was ranked fourth wealthiest country in the world behind Switzerland, the US, and Hong Kong [25]. From this prism, Australia's 'stingy' welfare system is a by-product of deliberate policy decisions to keep income-support payments low [26]. Unemployment benefits, in particular, have been indexed to the consumer price index rather than the cost of living or wage growth since 1994 [27].

Driven by inadequate government income support, Australians living in poverty often rely on charity to subsidize their basic needs [28], as eligibility for charity and government support are independent. As a result, the sector has expanded rapidly, as reflected by growing numbers of charity organizations and amounts of government funding allocated to them. Between 2014 and 2017, the number of charities delivering social services more than doubled and annual government funding to these organizations rose from AUD$4.5 billion to AUD$7.3 billion [29]. In March 2020, after the onset of the COVID-19 pandemic, charities providing emergency relief received an additional budget allocation of AUD$200 million to be spent over three financial years from 2020 to 2022 [30].

Further, recent estimates indicate that nearly half of all people requesting emergency relief from charity organizations presented four or more times within a 6-month period [31]. Critically, 95% of these individuals are also recipients of government income-support payments [32]. Overall, this suggests that the gap that institutionalized charity fills in the lives of the poor is generated by the inadequacy of government payments. In other words, the amount of government support is likely insufficient to meet people's needs, or else there would be no need for individuals to access this form of charity.

Prior to the present study, however, this proposition remained untested. The few existing, empirical studies focusing on welfare and charity have chiefly examined the relationship from the supply side. Such studies have considered whether increasing government support 'crowds out' charitable giving. For example, using historical data from England, Boberg-Fazlić and Sharp [33] showed that, contrary to expectations, areas that received greater public funding also enjoyed higher levels of charitable income. The relationship between the level of welfare support and the demand for charity is, however, far less understood. Scholars in the UK [2] and Australia [28] have previously argued that the rise in food banks is a direct result of reductions in welfare spending by governments. Consistent with this tenets, recent empirical evidence from the US [34] and Australia [35] has shown that increases in welfare payments during the COVID-19 pandemic enabled recipients to purchase food and other basic necessities—items which they may have otherwise sourced from charity. Given the paucity of empirical evidence in this space, our study makes an important contribution by explicitly linking the level of income support payments to the demand for charity using administrative data from the two largest charitable organizations in Australia.

## Shifts in Australian welfare policy in the wake of COVID-19

As in other advanced welfare states, the COVID-19 pandemic triggered an unprecedented number of temporary reforms in Australian welfare policy [36]. These reforms were remarkable within a global context and represented a complete reversal in Australian welfare policy—from 'punitive and targeted' to 'generous and comprehensive' [20]. Here, we focus on the

main payments directly impacting the amount of income support provided to welfare recipients.

In 2020, the Federal government launched two economic stimulus packages with the main aim of supporting people on welfare and designed to supplement existing income-support payments [37]. The first package was announced on 12 March 2020 and paid on 31 March 2020. It consisted of a lumpsum (i.e., one off) AUD$750 Economic Support Payment (ESP)—nearly as much as the 2020 weekly minimum wage (AUD$754)—to income-support recipients.

The second package, announced on 22 March 2020, included the Coronavirus Supplement (details below) and a second AUD$750 ESP paid on 13 July 2020 to those who were ineligible for the Supplement. In addition, two other lumpsum payments amounting to AUD$250 each were issued on 30 November 2020 and 1 March 2021 to those who were ineligible for the Supplement. Unlike the four lumpsum payments, the Coronavirus Supplement was paid at regular fortnightly intervals between April 2020 and March 2021. From 27 April to 24 September 2020, the Supplement was set at AUD$550 per fortnight. It was subsequently reduced to AUD$250 per fortnight from 25 September to 31 December 2020, and further reduced to AUD$150 per fortnight from 1 January to 31 March 2021.

In 2020 alone, more than 4 million people—or 19% of the population aged 15 and above—received the Coronavirus Supplement, while more than 7 million people—or 33% of the population aged 15 and above—received an ESP [38]. *S3 Fig in S1 File* provides a timeline of these payments and other related events, while *S1 Table in S1 File* outlines the eligibility criteria for each payment.

Altogether, these stimulus packages boosted the incomes of a significant share of the population, albeit temporarily. The group that benefitted the most were the unemployed, whose incomes increased from AUD$570 to AUD$1,115 per fortnight [39]. This near-doubling of unemployment benefits drastically reduced the rate of relative poverty among this group from 67% to 6.8% [39]. That is, the payments not only prevented poverty from worsening due to the impacts of COVID-19, but in fact reduced pre-pandemic poverty rates [39]. Accordingly, we expect that the payments also reduced the demand for institutionalized charity. In the next section, we present the results from our three-part empirical analyses.

## Results

### The demand for charity dropped markedly in March 2020

In this section, we describe aggregate trends in charitable assistance in the form of emergency relief provided by SVdP and TSA. *Fig 1* illustrates the combined number of assistance records from these two organizations from 1 January 2018 to 15 October 2021. To ensure that the trends are not driven by a specific charity, we compared the growth rates in assistance records for the two organizations. *S4 Fig in S1 File* shows remarkably similar trends for both charities from 2019 to 2020, including the sharp drop in assistance records at the end of March 2020, followed by a strong rebound in the following month. *S5 Fig in S1 File* also shows similar growth rates in assistance records for the two organizations for the 2020 to 2021 period, including the large spike in April 2021. To facilitate visualization, this figure aggregates the daily records into weekly data.

Of the three years with full annual data (i.e., 2018, 2019 and 2020), the demand for charity was lowest in 2020—when the new payments were introduced. Compared to an average of 143,519 assistance records between 2018 and 2019, there were only 98,803 records in 2020—a reduction of approximately 31.2%. Further, the decline in the demand for charity did not occur gradually. Rather, it began with a sudden drop during the last two weeks of March. This

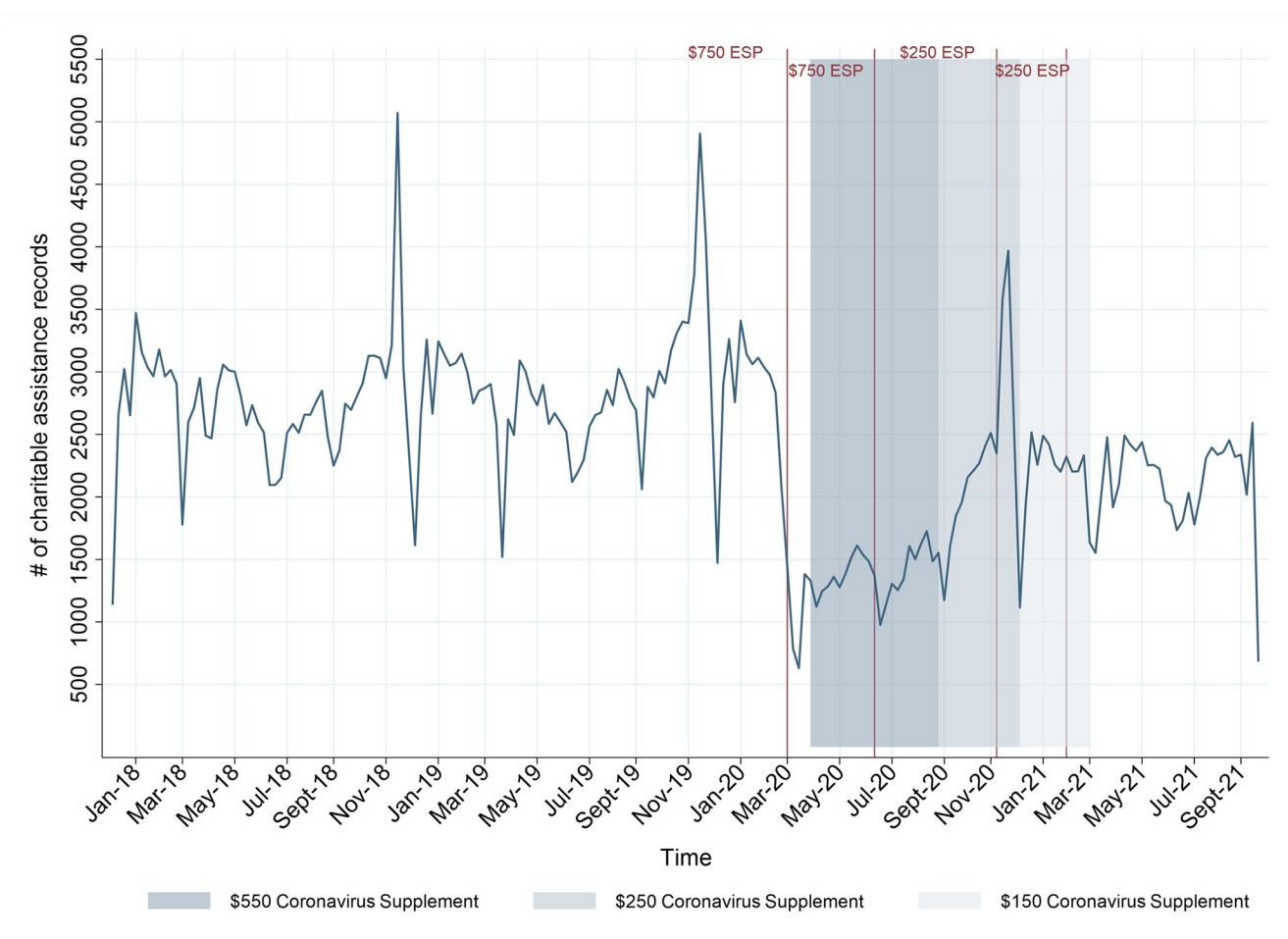

**Fig 1. Number of weekly assistance records (2018–2021).** This figure combines the number of weekly assistance records from both charities. Amounts are in Australian Dollars (AUD). ESP–Economic Support Payment.

coincides with the nationwide shutdown and the first AUD$750 ESP on 31 March 2020, represented by the first reference line within the graph.

The potential impact of the Coronavirus Supplement on the demand for charity can be gleaned from the shaded areas in *Fig 1*, where darker shades depict higher levels of the Supplement. From 27 April to 24 September 2020, when the Supplement was set at AUD$550 per fortnight, there were a total of 30,287 assistance records—46.9% less than the average number of records for the same period in 2018/2019. From 25 September to 31 December 2020, when the Supplement was at AUD$250 per fortnight, there were 29,435 assistance records—27.4% less than the equivalent period in 2018/2019. In the final phase of the Supplement, which lasted from 1 January to 31 March 2021, there were 27,780 assistance records—23.2% less than the 2018 and 2019 average. Altogether, these figures reveal a clear and meaningful pattern: when the Supplement was at its highest level, charitable assistance was at its lowest; and when the Supplement was gradually decreased, the demand for charity bounced back.

The lumpsum payments were also associated with reductions in the demand for charity, albeit only in the short-term. Of the four ESPs, the first AUD$750 payment at the end of March 2020 coincided with the largest decrease in charitable assistance. In the two weeks in

which the payment was issued, there were 1,613 assistance records—a reduction of 70.5% compared to the 2018/2019 average within the same period. There was a similar, though smaller, reduction in charity, of 45.8%, in July 2020 when the second AUD$750 ESP was issued. The third and fourth payments of AUD$250 in November 2020 and March 2021 were associated with smaller reductions in charitable assistance, amounting to 22.1% and 23.4%, respectively.

Overall, these results suggest a strong, negative association between the Coronavirus Supplement and ESPs and the demand for charity. However, this demand is also influenced by other factors unrelated to these two payments, including the 2020 nationwide shutdown, the 2021 localized Queensland lockdowns, seasonality, and other time-related factors (e.g., holidays). For example, Fig 1 shows that charitable assistance peaks in December of each year. This pattern is consistent with findings from earlier studies documenting that the volume of charitable giving is highest towards the end of the calendar year, coinciding with dates of religious significance (e.g., Christmas) [40]. There were also significant dips during key holidays, particularly in March or April (Easter) and October (Queen's Birthday). It follows that isolating the impact of the Coronavirus Supplement and ESPs on charitable assistance requires a modelling approach that can account for these factors.

## The coronavirus supplement and first economic support payment reduced charitable assistance

*Table 1* presents abridged results of difference-in-difference (DID) regressions using years 2018 and 2019 as the counterfactual (see *S2 Table in S1 File* for full results). The DID coefficients on the AUD$550 and AUD$250 Coronavirus Supplement were both negative and statistically significant across the three specifications, suggesting that the first two phases of the Supplement effectively reduced the demand for charity. The AUD$550 Supplement reduced assistance by 178 records per day (equivalent to a 47.0% reduction in the average daily number of assistance records in 2018/19), whereas the AUD$250 Supplement reduced assistance by 112 records per day (or a 27.0% reduction in records during the same period in 2018/2019). The AUD$150 Supplement was negatively and statistically significantly related to charitable assistance in two of the three specifications. Overall, an additional AUD$150 per fortnight reduced the demand for charity by 31 records per day (or a 7.8% decline in records during the same period in 2018/2019). Given the discrete nature of assistance records, we re-estimated Equation 1 using a Poisson regression model, with very similar results (see *S3 Table in S1 File*). As a robustness check, we also estimated a panel fixed effects regression model with daily assistance records as the unit of analysis. As shown in S4 Table in S1 File, the results were virtually identical to those of our main models.

Out of the four lumpsum payments, only the first AUD$750 ESP in March 2020 was statistically significantly associated with the demand for charity. Based on estimates from the combined data, it reduced assistance by 104 records per day over the two-week payment period (equivalent to a 28.6% reduction of the daily records during the same period in 2018/2019). In contrast, the second, third, and fourth ESPs did not significantly reduce charitable assistance. As we discuss later, this may be attributed to factors such as the smaller number of individuals who received those payments and their lower amount. We also examined the immediate impact of the first ESP on the demand for charity through a regression discontinuity analysis of assistance provided to welfare recipients using TSA data. The results highlight a sharp drop in charitable assistance in the first few days following the payment (*S6 Fig* and *S5 Table in S1 File*). This indicates that the events preceding the payment date (e.g., the nationwide shutdown and announcement of stimulus packages) were not responsible for the observed reductions in the demand for charity.

**Table 1. Unstandardized coefficients from difference-in-difference regressions of daily assistance records.**

|  | β (Pooled) | β (SVdP) | β (TSA) |
|---|---|---|---|
| 2020×CS1 | −177.75*** | −139.51*** | −47.47*** |
|  | (16.37) | (14.87) | (7.49) |
| 2020×CS2 | −111.61*** | −90.65*** | −29.10*** |
|  | (19.67) | (17.61) | (8.05) |
| 2021×CS3 | −31.49* | −37.53** | 0.30 |
|  | (14.42) | (12.46) | (6.40) |
| 2020×ESP1 | −104.42* | −96.81* | −24.39† |
|  | (50.40) | (42.92) | (13.66) |
| 2020×ESP2 | 16.53 | 5.95 | 12.05 |
|  | (29.90) | (25.78) | (12.22) |
| 2020×ESP3 | −29.69 | −32.96 | 15.42 |
|  | (37.44) | (37.13) | (12.65) |
| 2021×ESP4 | −18.02 | −30.61† | 19.47** |
|  | (18.93) | (16.62) | (7.30) |
| Shutdown controls | Yes | Yes | Yes |
| Lockdown controls | Yes | Yes | Yes |
| Holiday controls | Yes | Yes | Yes |
| Day-of-the-week effects | Yes | Yes | Yes |
| Day-of-the-year effects | Yes | Yes | Yes |
| Year effects | Yes | Yes | Yes |
| N | 1,384 | 1,384 | 949 |
| $R^2$ | 0.90 | 0.87 | 0.77 |

Clustered standard errors in parentheses. Statistical significance (two-sided tests)

†$p<0.10$

*$p<0.05$

**$p<0.01$

***$p<0.001$

SVdP–St. Vincent de Paul Society Queensland; TSA–The Salvation Army; CS1 –AUD$550 Coronavirus Supplement; CS2 –AUD$250 Coronavirus Supplement; CS3 –AUD$150 Coronavirus Supplement; ESP1 – 1st AUD$750 Economic Support Payment; ESP2 – 2nd AUD$750 Economic Support Payment; ESP3 – 3rd AUD$250 Economic Support Payment; ESP4 – 4th AUD$250 Economic Support Payment

## The demand for charity declined for eligible groups, but not for ineligible groups

In this section, we present the results of a second set of regression analyses of data from TSA, distinguishing between groups who were eligible and ineligible for the Coronavirus Supplement and the ESPs. To accomplish this, we estimated a triple difference model, as shown in Equation 2. Abridged results from this model are presented in *Table 2* (*S6 Table in S1 File* shows full results).

The coefficients on the three-way interaction terms for the AUD$550 and AUD$250 Coronavirus Supplement were negative and statistically significant, indicating that these payments reduced the demand for charity. With the AUD$550 Supplement, the demand for charity declined by nearly 22 records (equivalent to a 52.9% reduction in assistance to eligible individuals compared to 2018/2019), whereas with the AUD$250 Supplement it declined by 19 records per day (equivalent to a 39.3% decrease in assistance). Further, ineligible groups were unaffected by the AUD$550 and AUD$250 Coronavirus Supplement, as shown by the small and statistically insignificant coefficients on the two-way interaction terms (see *S6 Table in S1 File*). Collectively,

**Table 2. Unstandardized coefficients from triple-difference regressions of daily assistance records using the salvation army data.**

|  | β |
|---|---|
| 2020×CS1×Eligible | −21.93*** |
|  | (3.78) |
| 2020×CS2×Eligible | −19.14*** |
|  | (4.71) |
| 2021×CS3×Eligible | −0.99 |
|  | (3.43) |
| 2020×ESP1×Eligible | −3.02 |
|  | (5.45) |
| 2020×ESP2×Eligible | −1.73 |
|  | (4.08) |
| 2020×ESP3×Eligible | −2.36 |
|  | (9.57) |
| 2021×ESP4×Eligible | 3.47 |
|  | (4.02) |
| Year-Payment interactions | Yes |
| Year-Eligible interactions | Yes |
| Payment-Eligible interactions | Yes |
| Shutdown controls | Yes |
| Lockdown controls | Yes |
| Holiday controls | Yes |
| Day-of-the-week effects | Yes |
| Day-of-the-year effects | Yes |
| Year effects | Yes |
| N | 2,682 |
| R$^2$ | 0.66 |

Clustered standard errors in parentheses. Statistical significance (two-sided tests)

†$p<0.10$

*$p<0.05$

**$p<0.01$

***$p<0.001$

CS1 –AUD$550 Coronavirus Supplement; CS2 –AUD$250 Coronavirus Supplement; CS3 –AUD$150 Coronavirus Supplement; ESP1 – 1st AUD$750 Economic Support Payment; ESP2 – 2nd AUD$750 Economic Support Payment; ESP3 – 3rd AUD$250 Economic Support Payment; ESP4 – 4th AUD$250 Economic Support Payment. *Eligible* refers to records from individuals who were eligible for the Coronavirus Supplement and Economic Support Payments.

these results rule out the possibility that the observed reductions in the demand for charity were driven by factors unrelated to the Coronavirus Supplement (e.g., supply-side factors).

On the other hand, the three-way-interaction coefficients on the AUD$150 Supplement as well as the four ESPs were not statistically significant. This suggests that these other payments did not differentially affect the demand for charity among eligible and ineligible groups, potentially due to their less generous and universal nature.

## Discussion

This study has generated first-hand evidence of a relationship between the generosity of government income-support payments and the demand for institutionalized charity in the form

of emergency relief. To accomplish this, we used Australia as an exemplar, leveraging the numerous—yet temporary—social-policy reforms taking place during the initial stages of the COVID-19 pandemic in this advanced welfare state as a natural experiment.

All of our analyses pointed towards the same empirical reality: more generous income-support payments are associated with significant reductions in the demand for charity. This indicates that the amount of government support provided is insufficient to meet people's needs. Raw trends in charitable assistance before and after the introduction of the new Coronavirus Supplement and ESPs suggested a 31% reduction in 2020 compared to 2018/2019. Importantly, this decline seemed to coincide with the dates in which the new payments were introduced, beginning in March 2020. Moving from this descriptive account to difference-in-difference estimation yielded consistent results. A first set of DID models revealed large and statistically significant reductions in charity resulting from the Coronavirus Supplements and the first ESP. Additionally, the results of triple-difference models indicated that the AUD$550 and AUD$250 Coronavirus Supplements reduced the demand for charity among eligible groups, but not among ineligible groups.

This inverse relationship between the demand for charity and the level of income-support payments demonstrates that, in advanced welfare states such as Australia, poverty and charity are intertwined with policy decisions about welfare-state provision [28]. Policies that keep government income-support payments low not only create and perpetuate poverty; they also force many citizens to rely on charity to access basic necessities and generate the need for an institutionalized charity sector. Yet, this situation is not irreversible. The reforms to the Australian income-support system following the onset of the COVID-19 pandemic demonstrate that the state is fully capable of increasing the generosity of its welfare system. By raising the level of income support, governments can move from providing 'band-aid' solutions to poverty to directly addressing its structural causes [41–43].

Our results also hint that it is not only the amount of additional income support that matters, but also the recurrency of payments. While the regular, fortnightly Coronavirus Supplements unequivocally reduced the demand for charity, there was little evidence that the four lumpsum payments made a difference. Indeed, across the DID and triple-difference models, the relevant parameters on the ESPs were only statistically significant in 1 of 8 cases. The failure of the ESPs to reduce the demand for charity may be due to the tighter eligibility criteria for these payments, resulting in a smaller number of recipients. However, it may also stem from their unpredictability and irregularity, which may have limited recipients' agency to save, plan ahead, and strategize in their financial decisions [44].

These results have important implications for policy. Overall, they strongly suggest that sustained increases in government income-support payments are urgently required to meet the basic needs of all citizens and reduce reliance on charity for survival. While certain political groups argue that income-support benefits for the unemployed may de-incentivise job-seeking, research studies suggest that the effect may be moderated by other factors, such as eligibility requirements [45], and may ultimately result in more positive labour-market outcomes for recipients [46]. The economic viability of permanently and substantially increasing income-support payments remains, however, a legitimate concern. Yet our results suggest that effectively reducing the demand for charity to meet basic needs can be accomplished without reaching the maximum level of additional support offered by the AUD$550 Coronavirus Supplement. Our broader DID estimates combining information from the two charities indicate that a fortnightly AUD$150 income-support supplement yielded only a marginal reduction of ~8% in the demand for charity —or 0.05 percentage points (pp) per dollar invested. Thereafter, we observed steeper reductions. Moving from a AUD$150 to a AUD$250 fortnightly supplement further decreases the demand for charity by 19 pp (from 8% to 27%), or an average of 0.15 pp per marginal dollar. Moving from a AUD$250 to a AUD$550 supplement, leads to an additional decrease of 20 pp (from 27% to

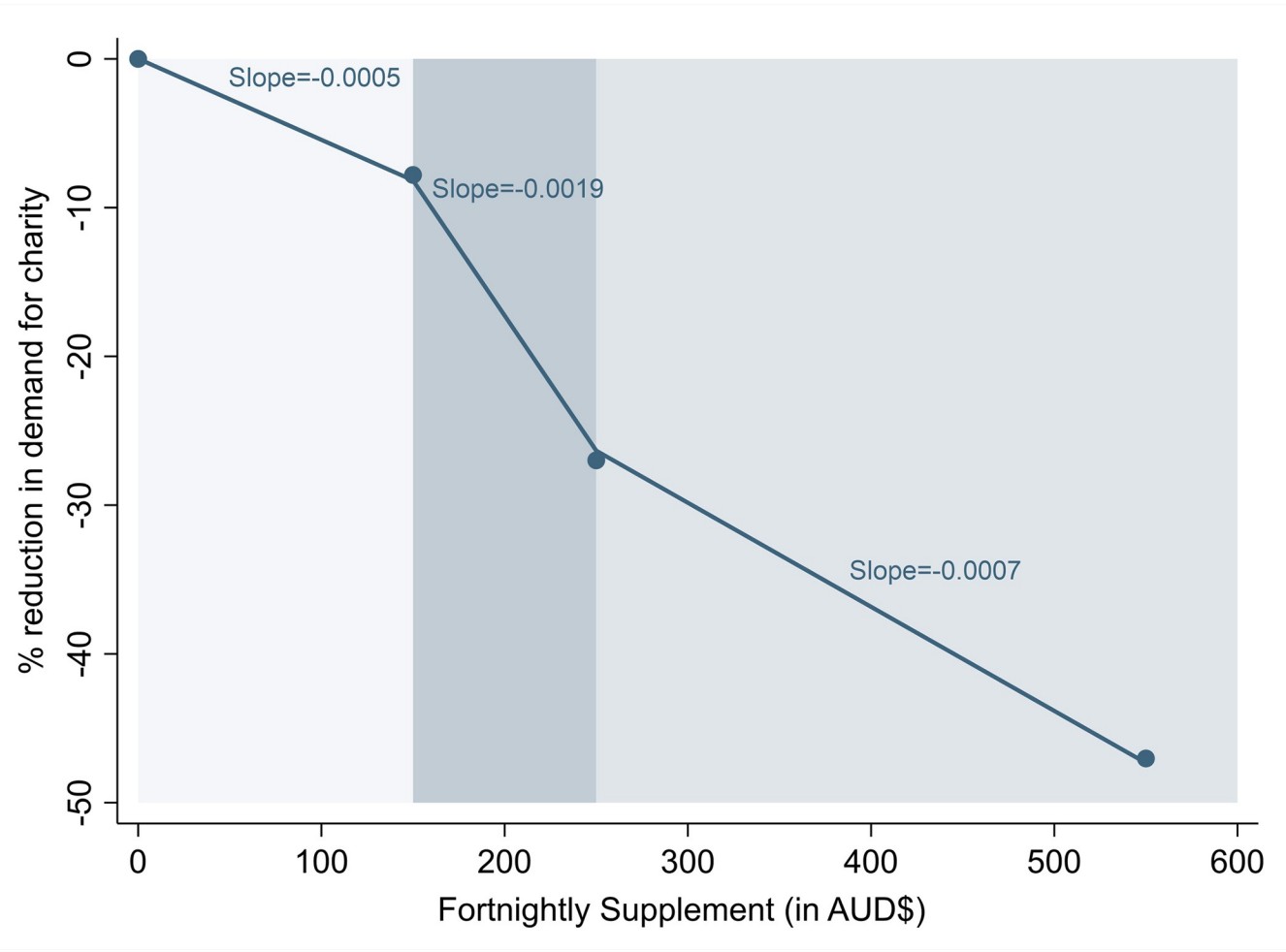

**Fig 2. Reductions in the demand for charity at different rates of the coronavirus supplement.** Blue dots represent the estimated reductions in the demand for charity at each rate of the Coronavirus Supplement, based on the results of difference-in-difference models in Table 1. Lines connecting each of the point estimates were added to visualize the predicted reduction in assistance records per additional dollar of the Supplement.

47%), but only of 0.07 pp per marginal dollar. These 'back-of-the-envelope' calculations—illustrated in Fig 2—point to the existence of a 'sweet spot' in the level of additional support that would represent the greatest return per dollar invested. Based on our models, this figure sits at approximately AUD$250 per fortnight (or AUD$18 per day).

## Study limitations and avenues for further research

Given the aggregate nature of the data, we were unable to distinguish charitable assistance by the type of income-support payment received (e.g., JobSeeker, Parenting Payment, Disability Support Payment, etc.). As a result, our estimates provide the average effect of the additional income-support payments on the need for charitable assistance.

Additionally, the data did not include information on the characteristics (e.g., age, gender, marital status) of individuals who accessed emergency relief. Future studies using more disaggregated data can therefore examine whether the effect of these additional payments differ across population groups and according to the type of welfare support received. Studies that extrapolate our findings to other country contexts are also warranted.

## Conclusion

For people in poverty who use charity to survive, increasing income support eliminates multiple barriers to obtaining goods and services that are basic and essential—including the stress, shame, and stigma associated with presenting to charity and having to demonstrate need [10]. It also enables them to access resources as a citizenry right [16], rather than as something akin to begging, maximizing their ability to exercise agency and live with dignity. For charitable organizations, the additional capacity freed up by increasing income support would allow them to offer better assistance to other populations that may fall through the cracks of the welfare system (e.g., refugees) and to redirect efforts to address the root causes of poverty rather than simply ameliorating its consequences. For the state, increasing the level of income support represents a viable alternative to assisting vulnerable populations compared to funding charity organizations to accomplish this task. Indeed, direct assistance from government to people in need may involve fairer, more transparent and more efficient processes than using charities as the 'middleman'.

Given these potential benefits, our results enable us to formulate thought-provoking, humanistic and aspirational propositions. Assuming that reducing the need for citizens to resort to charity to eat, pay rent, or dress their children is indeed a policy goal, our findings indicate that *halving the demand for charity requires raising pre-pandemic income-support payments by AUD\$42 per day*. Less ambitious reductions of 25% and 10% would require lifting pre-pandemic income-support payments by AUD\$17 and AUD\$12, respectively. While these figures apply to the contemporary Australian society, the underlying relationships between government support and institutionalized charity likely extend to other advanced welfare states. Further research replicating our approach and methods across institutional and temporal contexts is warranted.

## Materials and methods

### Data

Ethics approval to conduct this study was issued by our institution's Human Research Ethics Committee. The cornerstone of our data are assistance records made available by the two largest providers of charitable assistance in the state of Queensland, Australia—the St. Vincent de Paul Society (SVdP) and The Salvation Army (TSA). These data are not publicly available and were secured thanks to long-running research partnerships between the research team and these two organizations. Queensland is the third largest Australian state in terms of population, with an estimated 5.2 million residents in 2021, or 20% of the Australian population [47]. Data from the Australian Government Department of Social Services show that SVdP and TSA jointly account for the majority of ER provision in Queensland, both in terms of the number of individuals assisted as well as the number of ER sessions administered [48]. Any individual is welcome to present at these charity organizations and request emergency-relief support. Between January and June 2021, the latest reporting period for which complete data are available, SVdP assisted 38,159 individuals (53% of the total), while TSA assisted 11,531 individuals (16% of the total) [48]. Over the same time period, SVdP administered ER 32,970 times (28% of the total), while TSA delivered ER 40,136 times (33% of the total) [48]. Combined, these two organizations account for approximately 69% of ER clients and 61% of all ER sessions in Queensland [48]. Across Australia, SVdP and TSA employ close to 12,000 paid staff (3,212 and 8,594, respectively) in addition to more than 56,000 volunteers (33,261 and 23,405) [49]. In 2020 alone, the two organizations had a combined revenue of approximately AUD\$1.5 billion (AUD\$550 million for SVdP and AUD\$979 million for TSA), of which nearly half came from government [49].

The databases maintained by SVdP and TSA contain detailed assistance records, including information on the date on which assistance was provided. From these, we built a time-series

dataset tracking the daily number of assistance records per charity from 1 January 2018 to 15 October 2021 ($n$ = 1,384 daily observations). This is our outcome variable of interest. We focus on the number of assistance records per day rather than the amount of assistance provided because the latter depends on organizational factors (e.g., budgetary constraints) and may not accurately reflect the level of need. Additionally, the value of some goods offered by charity organizations as emergency relief cannot be monetized without making arbitrary assumptions (e.g., second-hand clothing or furnishings). The granularity and timeframe of our data allow us to link the daily volume of charitable-assistance provision to the precise payment dates, offering unparalleled insights into how the demand for institutionalized charity was impacted by changes in government income support.

The dates of the Coronavirus Supplement and ESPs are our key explanatory variables. To characterize the different payments, we use a series of dummy variables identifying the dates in which the payments were in place. The dummy variables take the value one for those dates, and the value zero on all other dates, *across the four years in the data*. For example, the dummy variable for the AUD$550 Coronavirus Supplement takes the value one for the period 27 April to 24 September in 2018, 2019, 2020 and 2021, and the value zero on all other dates. For the lumpsum ESPs, we use a two-week window as the payment period, given that these payments were fully disbursed to eligible recipients within that timeframe. For example, the dummy variable for the first AUD$750 ESP takes the value one from 31 March to 14 April in 2018, 2019, 2020 and 2021 and the value zero on all other dates. A comprehensive list of dates in which the payment variables take the value one is presented in *S7 Table in S1 File*.

## Analytic strategy

Our analytic strategy is divided into three parts.

**Raw trends.** In Part 1, we describe the overall trends in the number of charitable assistance records for the entire observation window spanning 1 January 2018 to 15 October 2021. The objective of this exercise is to gauge whether the period coinciding with the payment of the Coronavirus Supplement and ESPs is associated with a lower volume of assistance records. These raw comparisons also allow us to measure the unconditional differences in the number of assistance records before and after each of the payments were introduced.

**Difference-in-differences estimation.** In Part 2, we estimate Difference-in-Differences (DID) models with multiple treatments and time periods. DID estimation enables the retrieval of the estimated effect of a treatment on an outcome by comparing differences in that outcome between groups that were and were not exposed to the treatment [50]. Our exact model builds on the approach used in other studies [51], and allows us to compare daily numbers of charitable assistance records in payment vs. non-payment dates in 2020 or 2021 (when payments were active) vs. 2018/2019 (when payments were not active). The model also controls for other events occurring around the payment periods (e.g., shutdown and lockdown dates) as well as seasonality (e.g., Christmas) and other time-related factors (e.g., public holidays). This ensures that changes in charitable assistance across dates are not driven by these other factors.

Our empirical model takes the following form:

$$
\begin{aligned}
Assistance_i = {} & \alpha + \beta_1(CS_{1i} \times Year_{2020i}) + \beta_2(CS_{2i} \times Year_{2020i}) + \beta_3(CS_{3i} \times Year_{2021i}) + \beta_4(ESP_{1i} \\
& \times Year_{2020i}) + \beta_5(ESP_{2i} \times Year_{2020i}) + \beta_6(ESP_{3i} \times Year_{2020i}) + \beta_7(ESP_{4i} \\
& \times Year_{2021i}) + \beta_8(Shutdown_i \times Year_{2020i}) + \beta_9(Lockdown_i \times Year_{2021i}) + \delta_i + \mu_i \\
& + \epsilon_i
\end{aligned}
$$

Here, *Assistance$_i$* represents an outcome variable capturing the daily number of assistance records. The model includes dummy variables for each of the payment periods ($CS_{1i}$, $CS_{2i}$,

$CS_{3i}$, $ESP_{1i}$, $ESP_{2i}$, $ESP_{3i}$, $ESP_{4i}$), years ($Year_{2020i}$ and $Year_{2021i}$), and nationwide shutdown ($Shutdown_i$) and Queensland lockdown ($Lockdown_i$) dates, as well as interactions between these. While the nationwide shutdown came into effect on 23 March 2020, the easing of restrictions took place in phases and varied across Australian states. Here, we assume that the shutdown lasted from 23 March 2020 to 1 May 2020 in Queensland. The end date is based on the first easing of restrictions, which took effect on 2 May 2020 and allowed Queenslanders to travel within 50km of their homes for recreation and other purposes. The coefficients on the interaction terms $Year_{2020} \times CS_1$, $Year_{2020} \times CS_2$, $Year_{2021} \times CS_3$, $Year_{2020} \times ESP_1$, $Year_{2020} \times ESP_2$, $Year_{2020} \times ESP_3$ and $Year_{2021} \times ESP_4$ are of key interest, and represent the DID estimates of the effects on charitable assistance of the Coronavirus Supplement and the ESPs. Vector $\delta$ encompasses the main effects of all payments, shutdown/lockdowns, and years, whereas vector $\mu$ captures controls for public holidays (e.g., New Year's Day, Australia Day, Christmas Day), day-of-the-week fixed effects, day-of-the-year fixed effects, and year fixed effects. To ensure that the results are not driven by a specific charity organization, we also estimate the model for SVdP and TSA separately. Across all models, standard errors are clustered at the day (i.e., 1 to 365) level.

To confirm that trends in charitable assistance are comparable prior to the onset of the COVID-19 pandemic, we plotted the combined number of weekly assistance records across the different years. S7 Fig *in S1 File* shows very similar patterns of weekly assistance between 2020 and the two previous years up until the third week of March 2020, when the nationwide shutdown came into effect. We also implemented a more rigorous test of the parallel trends assumption by estimating differences in the number of weekly assistance records before and after the nationwide shutdown. S8 Fig in S1 File plots the coefficients from a fixed effects regression with leads and lags for the date of the nationwide shutdown (i.e., at t = 0). The estimated coefficients prior to the shutdown are near-zero and not statistically significant. In other words, there were no substantive differences in assistance between 2020 and previous years up until the nationwide shutdown.

**Triple difference estimation.** While the vast majority of individuals accessing charity in the form of emergency relief are also welfare recipients, the data also contain a small share of records from individuals who are not eligible for income-support payments and, as a result, for the Coronavirus Supplement and the ESPs. Because only citizens and permanent residents receiving government income support are eligible to receive such payments (see *S1 Table in S1 File*), other groups such as international students, temporary migrants, and asylum seekers were not entitled to these payments. In Part 3 of the analyses, we exploit these eligibility differences to estimate a triple difference model. This specification compares assistance provided to people who were eligible and ineligible in payment and non-payment dates across the years. A reduction in the demand for charity among eligible groups, but not among ineligible groups, would further support the argument that it was indeed the Coronavirus Supplement and ESPs that reduced the demand for charitable assistance (instead of other factors, such as changes in supply).

To establish this, we use TSA data, as it is not possible to disaggregate assistance records by eligibility status in the SVdP data. The dataset used for this analysis follows the same time-series format as the one used for the previous analysis, but it includes two sets of daily records: (i) records capturing assistance to individuals eligible for the Coronavirus Supplement and ESPs, and (ii) records capturing assistance to ineligible individuals. To distinguish between the two types of records, we create a new *Eligible* dummy variable—where value one denotes assistance to eligible individuals and value zero denotes assistance to ineligible individuals.

Our empirical model is as follows:

$$
\begin{aligned}
Assistance_i = {} & \alpha + \beta_1(CS_{1i} \times Year_{2020i} \times Eligible) + \beta_2(CS_{2i} \times Year_{2020i} \times Eligible) + \beta_3(CS_{3i} \\
& \times Year_{2021i} \times Eligible) + \beta_4(ESP_{1i} \times Year_{2020i} \times Eligible) + \beta_5(ESP_{2i} \times Year_{2020i} \\
& \times Eligible) + \beta_6(ESP_{3i} \times Year_{2020i} \times Eligible) + \beta_7(ESP_{4i} \times Year_{2021i} \times Eligible) \\
& + \beta_8(Shutdown_i \times Year_{2020i} \times Eligible) + \beta_9(Lockdown_i \times Year_{2021i} \times Eligible) \\
& + \delta_i + \mu_i + \epsilon_i
\end{aligned}
$$

The model in Equation 2 includes three-way interactions between the year dummy variables (i.e., $Year_{2020i}$ and $Year_{2021i}$), the payment dummy variables (i.e., $CS_1$, $CS_2$, $CS_3$, $ESP_1$, $ESP_2$, $ESP_3$, $ESP_4$), and the new *Eligible* dummy variable. The coefficients on the three-way-interaction terms (e.g., $Year_{2020} \times CS_1 \times Eligible$) are of key interest and denote the estimated effects of the payments. This model also includes each of the main effects and two-way interactions (vector $\delta$) and the same controls and fixed effects as the first model (vector $\mu$).

## Supporting information

**S1 File.**
(DOCX)

**S2 File.**
(PDF)

**S1 Data.**
(XLS)

**S2 Data.**
(XLS)

## Acknowledgments

We are grateful to St. Vincent de Paul Society Queensland and The Salvation Army for sharing their data and providing operational support. We are also indebted to Dr. Johana Susanto and Vaughan Olliffe for providing important advice on how to approach the Salvation Army data.

## Author Contributions

**Conceptualization:** Christine Ablaza, Francisco Perales, Cameron Parsell, Nathan Middlebrook, Richard N. S. Robinson, Ella Kuskoff, Stefanie Plage.

**Data curation:** Christine Ablaza, Nathan Middlebrook.

**Formal analysis:** Christine Ablaza, Francisco Perales.

**Investigation:** Francisco Perales, Cameron Parsell.

**Supervision:** Cameron Parsell.

**Validation:** Nathan Middlebrook, Richard N. S. Robinson, Stefanie Plage.

**Visualization:** Ella Kuskoff.

**Writing – original draft:** Christine Ablaza.

**Writing – review & editing:** Francisco Perales, Cameron Parsell, Nathan Middlebrook, Richard N. S. Robinson, Ella Kuskoff, Stefanie Plage.

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
