## [Decision Letter · Decision Letter 0]

4 Jan 2023

PONE-D-22-32871Increases in income-support payments reduce the demand for charityPLOS ONE

Dear Dr. Ablaza,

Thank you for submitting your manuscript to PLOS ONE. After careful consideration, we feel that it has merit but does not fully meet PLOS ONE’s publication criteria as it currently stands. Therefore, we invite you to submit a revised version of the manuscript that addresses the points raised during the review process.

Please consider the suggestions especially on the following areas:1. Literature review is weak.

2. problem statement is vague.

3. methodological explanations are required.

We look forward to receiving your revised manuscript.

Kind regards,

Muhammad Khalid Bashir, PhD

Academic Editor

PLOS ONE

Journal Requirements:

"This research was partially supported by the Australian Research Council Centre of Excellence for Children and Families over the Life Course (project number CE200100025), an Australian Future Fellowship Research Grant (FT180100250), https://www.arc.gov.au/  and the St. Vincent de Paul Society Queensland, https://qld.vinnies.org.au/. The funders had no role in study design, data collection and analysis, decision to publish, or preparation of the manuscript."

"I have read the journal's policy and the authors of this manuscript have the following competing interests: N.M. works as Community Engagement Manager at St. Vincent de Paul Society Queensland (whose data was used in this study). The University of Queensland and St. Vincent de Paul Society Queensland have an ongoing research partnership, but the Society was not involved in the conceptualization, analysis, and preparation of this manuscript. The views expressed in this article do not necessarily represent the views of St. Vincent de Paul Society or The Salvation Army."

Please confirm that this does not alter your adherence to all PLOS ONE policies on sharing data and materials, by including the following statement: ""This does not alter our adherence to  PLOS ONE policies on sharing data and materials.” (as detailed online in our guide for authors http://journals.plos.org/plosone/s/competing-interests).  If there are restrictions on sharing of data and/or materials, please state these. 

Please note that we cannot proceed with consideration of your article until this information has been declared. 

**Additional Editor Comments:**

Please consider the suggestions of the reviewers while revising your paper especially in:

1. Literature review is weak.

2. problem statement is vague.

3. methodological explanations are required.

Reviewers' comments:

Reviewer's Responses to Questions

**Comments to the Author**

1. Is the manuscript technically sound, and do the data support the conclusions?

Reviewer #1: Partly

Reviewer #2: Yes

2. Has the statistical analysis been performed appropriately and rigorously? 

Reviewer #1: No

Reviewer #2: Yes

3. Have the authors made all data underlying the findings in their manuscript fully available?

Reviewer #1: No

Reviewer #2: No

4. Is the manuscript presented in an intelligible fashion and written in standard English?

Reviewer #1: Yes

Reviewer #2: Yes

5. Review Comments to the Author

Reviewer #1: the paper objective is to investigate if support payments are associated with reductions in charity demand

- the authors need to provide literature review on the topic

- the amount of government support, is it coined with the level of charity, are those who get charity eligible for government support

- what are the characteristics of the sample? is the charity sample the same as the sample that gets government support?

- the authors used count data for the dependent variable and did not use Poisson regression, would this model improve their results? is the frequency for government support increased? is the quantity of support meets the minimum needs of the poor in terms of poverty line?

Reviewer #2: I would like to appreciate the authors for their kind effort to conduct this study entitled “ Increases in income-support payments reduce the demand for charity”. It is an interesting topic which may contribute majorly toward the country’s welfare policy development. The current version has some flaws and in regard of this I have suggested some improvements.

Title: Title is so simple it can be improved based on the study.

Abstract: Authors should incorporate the methodology in abstract

Keywords: Instead of COVID-19 and Administrative data, there should be more suitable keywords. Authors should replace them.

Introduction:

A clear research problem statement is missing.

Line 107-113: I think in this one paragraph authors have successfully completed their research, as they mentioned the methodology and their findings. If it is, then what does remaining pages present? Authors should consider this, and they should provide their research questions, hypothesis.

Author should also discuss the side-effects of unemployment benefits on the country’s economy, briefly explain sustainability of this system. Is it a attractive policy in term of unemployment?

S3: first economic support payment was delivered from 31 March 2020, followed by Coronavirus supplements, authors should mention the time period for which this support was provided? When it is ended or still it is ongoing.

Line 183-199. Authors should explain why only these two organisations were selected and what was their selection criteria? Their size, their number of personals etc. Moreover, in the text authors mentioned moths but in figure (S4) and (S5) authors used number of weeks, it would be better to use same. I suggest to use months also in figure. Similarly, the information about the average amount these two organisations are providing should be provided. Similarly, how many Australians were supported by them per year or month before and during and after the severe impacts of Pandemic e.g. in new normal.

Figure 1: in each year, the charity is on its peak in month of November as compare to the other months even though during pandemic. Author should explain why does this trend exist? And why the charity has been reduced in March 2020 and similarly in November 2020 it turned to up???

Line 267-272: eligible and ineligible persons for ESP and Coronavirus Supplement; did a person receive both types of support and charity from any of two organisations? If no, then why they not received because during the pandemic each and every person seemed like a deserving person? Kindly elaborate briefly.

Methodology section is at the end, it should be before the Result section.

Conclusion is missing

Policy implications should be incorporated.

The limitations of the study, and future scope of the study should be incorporated.

Best of Luck

6. PLOS authors have the option to publish the peer review history of their article (what does this mean?). If published, this will include your full peer review and any attached files.

Reviewer #1: No

Reviewer #2: No

---

## [Author Response · Author response to Decision Letter 0]

12 Apr 2023

Dear Prof Muhammad Khalid Bashir,

Thank you for inviting us to revise and resubmit our research paper “Increases in income-support payments reduce the demand for charity: A difference-in-difference analysis of charitable-assistance data from Australia over the COVID-19 pandemic” for possible publication in PLoS One. We are grateful for the helpful and constructive feedback issued by the Reviewers, and we have worked towards incorporating it into our manuscript. 

As recommended by the Editor, we have concentrated on enhancing the literature review; refining and making explicit the problem statement; and providing necessary methodological clarifications. As detailed in our responses, we have also amended our Financial Disclosure as follows: “This research was supported by (i) the Australian Research Council Centre of Excellence for Children and Families over the Life Course (project number CE200100025), (ii) an Australian Future Fellowship Research Grant FT180100250), https://www.arc.gov.au/ and (iii) the St. Vincent de Paul Society Queensland, https://qld.vinnies.org.au/. There was no additional external funding received for this study. The funders had no role in study design, data collection and analysis, decision to publish, or preparation of the manuscript.”

Overall, we believe that the changes made have significantly improved the paper. We hope that you consider this revised manuscript positively and look forward to hearing from you.

With best wishes,

The authors

REVIEWER 1

Comment: The authors need to provide literature review on the topic

Response: Thank you for this suggestion. To our knowledge, there are no studies addressing this precise research topic – which makes our contribution particularly novel. However, following this Reviewer’s advice, we have now summarised the findings of those studies that are closest to ours (p.6).

Comment: The amount of government support, is it coined with the level of charity, are those who get charity eligible for government support

Response: Thank you. We now remind readers that eligibility for the receipt of charity and eligibility to government support are fully independent (p.5).

Comment: What are the characteristics of the sample? is the charity sample the same as the sample that gets government support?

Response: Thank you. The unit of analysis in this study is the overall level of payments within a day – not the individual. This is explained in p.16. As such, there is no possibility to examine individual characteristics within our data.

Comment: The authors used count data for the dependent variable and did not use Poisson regression, would this model improve their results? 

Response: Thank you for this insightful methodological observation. To our knowledge, count models (including Poisson regression) are preferrable when dealing with small counts and skewed distributions – none of this is a major issue in our data. However, as we explain in the paper, we replicated our models using Poisson regression and obtained similar results (see p.10).

Comment: Is the frequency for government support increased? is the quantity of support meets the minimum needs of the poor in terms of poverty line?

Response: Thank you for this observation. A key motivation behind this paper is to demonstrate that the amount of government support is insufficient to meet need. Else, there would be no need for individuals to access charity to meet basic needs. We have made this goal even more explicit in the revised manuscript (see p.5 and p.13).

REVIEWER 2

Comment: Title: Title is so simple it can be improved based on the study.

Response: Thank you for this suggestion. We have expanded the title, as follows: “Increases in income-support payments reduce the demand for charity: A difference-in-difference analysis of charitable-assistance data from Australia over the COVID-19 pandemic”.

Comment: Abstract: Authors should incorporate the methodology in abstract

Response: Thank you. The abstract now reads: “We draw on this natural experiment and time-series data from the two largest charity organizations in Queensland, Australia to examine how these payments altered the demand for institutionalized charity. We model these data using difference-in-difference regression models to approximate causal effects” (p.2).

Comment: Keywords: Instead of COVID-19 and Administrative data, there should be more suitable keywords. Authors should replace them.

Response: Thank you for this suggestion. We have checked papers recently published in PLoS One (see https://journals.plos.org/plosone/) and it seems that keywords are not publicly shown. Therefore, there was no need to alter the keywords at this stage.

Comment: Introduction: A clear research problem statement is missing.

Response: Thank you. We agree with this recommendation and have added a clearer problem statement in the introduction (p.4).

Comment: Line 107-113: I think in this one paragraph authors have successfully completed their research, as they mentioned the methodology and their findings. If it is, then what does remaining pages present? 

Response: Thanks for this observation. We are not completely sure we understood this comment. The Reviewer may be referring to our inclusion of a preview of the findings at the end of the introduction. We appreciate this is not the norm across all disciplines, although it isn’t unusual in our discipline of sociology. In any case, we have gone ahead and removed this preview of findings (see p.4).

Comment: Author should also discuss the side-effects of unemployment benefits on the country’s economy, briefly explain sustainability of this system. Is it a attractive policy in term of unemployment?

Response: Thank you. Again, we are not sure whether we understood this comment correctly. We have nevertheless added a sentence stating that “While certain political groups argue that income-support benefits for the unemployed may de-incentivise job-seeking, research studies suggest that the effect may be moderated by other factors, such as eligibility requirements [45], and ultimately result in positive labour-market outcomes for recipients [46]” (p.14). A full discussion of the pros and cons of unemployment benefits as a policy solution is out of scope for the current study and can be found elsewhere. 

Comment: S3: first economic support payment was delivered from 31 March 2020, followed by Coronavirus supplements, authors should mention the time period for which this support was provided? When it is ended or still it is ongoing.

Response: Thank you for this comment. As we mention in the paper, this is a lumpsum (i.e., one off) payment. The relevant section now reads: “The first package was announced on 12 March 2020 and paid on 31 March 2020. It consisted of a lumpsum (i.e., one off) AUD$750 Economic Support Payment (ESP)—nearly as much as the 2020 weekly minimum wage (AUD$754)—to income-support recipients” (p.6). We have added the words “(i.e., one off)” to make this clearer.

Comment: Line 183-199. Authors should explain why only these two organisations were selected and what was their selection criteria? Their size, their number of personals etc. 

Response: Thank you. We have expounded on our reasons for selecting the two organisations in the Materials and Methods section, including providing supporting evidence on the volume of assistance provided, clients assisted, number of personnel, and total revenue (pp.16-17). 

Comment: Similarly, the information about the average amount these two organisations are providing should be provided.

Response: Thank you. As we note in the paper, it is not possible to monetize all of the help provided by these organizations: “We focus on the number of assistance records per day rather than the amount of assistance provided because the latter depends on organizational factors (e.g., budgetary constraints) and may not accurately reflect the level of need. Additionally, the value of some goods offered by charity organizations as emergency relief cannot be monetized without making arbitrary assumptions (e.g., second-hand clothing or furnishings)” (p.17). Therefore, we are unable to implement this suggestion.

Comment: Moreover, in the text authors mentioned moths but in figure (S4) and (S5) authors used number of weeks, it would be better to use same. I suggest to use months also in figure.

Response: Thank you for the suggestion on using “months” in the figure axes. We have amended this for all relevant supplementary figures.

Comment: Similarly, how many Australians were supported by them per year or month before and during and after the severe impacts of Pandemic e.g. in new normal.

Response: Thank you. Figure S7 showcases the number of Australians supported by the two organizations over the complete observation period in the data available to the research team—which is the period of analysis. We do not hold data for earlier or posterior time frames.

Comment: Figure 1: in each year, the charity is on its peak in month of November as compare to the other months even though during pandemic. Author should explain why does this trend exist? And why the charity has been reduced in March2020 and similarly in November 2020 it turned to up???

Response: Thank you for this insightful observation. Charitable assistance does peak in December of each year, as shown in Figure 1. This is consistent with the findings of earlier studies documenting that charitable giving peaks in December, coinciding with dates of religious significance and the Christmas holidays. We now raise the importance of accounting for seasonality in charitable assistance in several sections of the document (p.9 and p.19). Regarding the second point, we discuss these trends within the manuscript (p.8). In particular, we argue that the drop in assistance in the last two weeks of March 2020 resulted from the nationwide shutdown and the first $750 ESP. 

Comment: Line 267-272: eligible and ineligible persons for ESP and Coronavirus Supplement; did a person receive both types of support and charity from any of two organisations? If no, then why they not received because during the pandemic each and every person seemed like a deserving person? Kindly elaborate briefly.

Response: Thank you. We are not sure we fully understand this point. We nevertheless now mention within the paper that eligibility to charity and government support are independent (p.5) and that any individual is welcome to present at a charity organization and request support (p.16). The precise eligibility for each of the government payments is outlined in detail in Table S1. We are unable to comment meaningfully on why the Australian government settled on these eligibility criteria.

Comment: Methodology section is at the end, it should be before the Result section.

Response: Thank you. We have consulted the author guidelines for PloS One and this specific journal allows for the ‘Materials and Methods’ section to be positioned at the end of the manuscript, after the results question: https://journals.plos.org/plosone/s/submission-guidelines#loc-manuscript-organization

Comment: Conclusion is missing.

Response: Thank you. We failed to include this subheading. It has now been added to the paper (see p.15).

Comment: Policy implications should be incorporated.

Response: Thank you. We have made our policy implications more explicit, as these were admittedly vaguely phrased in the initial submission (see p.14).

Comment: The limitations of the study, and future scope of the study should be incorporated.

Response: Thank you. We have added a paragraph at the end covering these points (see p.15).

COMMENTS FROM THE EDITOR

Comment: Please ensure that your manuscript meets PLOS ONE's style requirements, including those for file naming.

Response: Thank you. We have examined the style requirements again and ensure, to the best of our ability, that the manuscript adheres to them.

Comment: 2. We note that the grant information you provided in the ‘Funding Information’ and ‘Financial Disclosure’ sections do not match. When you resubmit, please ensure that you provide the correct grant numbers for the awards you received for your study in the ‘Funding Information’ section.

Response: Thank you. We have ensured the wording of these two sections is accurate and consistent with each other.

Comment: 3. Thank you for stating in your Funding Statement: "…”. Please provide an amended statement that declares *all* the funding or sources of support (whether external or internal to your organization) received during this study, as detailed online in our guide for authors at http://journals.plos.org/plosone/s/submit-now. Please also include the statement “There was no additional external funding received for this study.” in your updated Funding Statement. Please include your amended Funding Statement within your cover letter.

Response: Thank you. We can confirm these are *all* of the funding sources. We have reworded this section to make it clearer, as well as adding the sentenced provided. The section now reads: 

“This research was supported by (i) the Australian Research Council Centre of Excellence for Children and Families over the Life Course (project number CE200100025), (ii) an Australian Future Fellowship Research Grant FT180100250), https://www.arc.gov.au/ and (iii) the St. Vincent de Paul Society Queensland, https://qld.vinnies.org.au/. There was no additional external funding received for this study. The funders had no role in study design, data collection and analysis, decision to publish, or preparation of the manuscript.”.

Comment: Thank you for stating the following in the Competing Interests section: "…". Please confirm that this does not alter your adherence to all PLOS ONE policies on sharing data and materials, by including the following statement: ""This does not alter our adherence to PLOS ONE policies on sharing data and materials.” If there are restrictions on sharing of data and/or materials, please state these. Please include your updated Competing Interests statement in your cover letter.

Response: Thank you. We have added the missing sentence to the Competing Interest statement: “We have read the journal's policy and the authors of this manuscript have the following competing interests to declare: N.M. works as Community Engagement Manager at St. Vincent de Paul Society Queensland (whose data was used in this study). The University of Queensland and St. Vincent de Paul Society Queensland have an ongoing research partnership, but the Society was not involved in the conceptualization, analysis, and preparation of this manuscript. The views expressed in this article do not necessarily represent the views of St. Vincent de Paul Society or The Salvation Army. This does not alter our adherence to PLOS ONE policies on sharing data and materials.”

Comment: PLOS only allows data to be available upon request if there are legal or ethical restrictions on sharing data publicly. […] In your revised cover letter, please address the following prompts: a) If there are ethical or legal restrictions on sharing a de-identified data set, please explain them in detail (e.g., data contain potentially sensitive information, data are owned by a third-party organization, etc.) and who has imposed them (e.g., an ethics committee). Please also provide contact information for a data access committee, ethics committee, or other institutional body to which data requests may be sent.

Response: Thank you. We have now obtained approval from both the St. Vincent de Paul Society Queensland and The Salvation Army to share the data used for analysis as part of our supplementary materials. Following their approval, we have now attached the raw data as Supplementary Information in our submission. In addition, we have also deposited the raw data in a Plos One-approved data repository – the UK Data Service – where it is being reviewed by the editor prior to making the dataset available for use.

---

## [Decision Letter · Decision Letter 1]

7 Jun 2023

Increases in income-support payments reduce the demand for charity: A difference-in-difference analysis of charitable-assistance data from Australia over the COVID-19 pandemic

PONE-D-22-32871R1

Dear Dr. Ablaza,

We’re pleased to inform you that your manuscript has been judged scientifically suitable for publication and will be formally accepted for publication once it meets all outstanding technical requirements.

Kind regards,

Muhammad Khalid Bashir, PhD

Academic Editor

PLOS ONE

Additional Editor Comments (optional):

Reviewers' comments:

Reviewer's Responses to Questions

**Comments to the Author**

1. If the authors have adequately addressed your comments raised in a previous round of review and you feel that this manuscript is now acceptable for publication, you may indicate that here to bypass the “Comments to the Author” section, enter your conflict of interest statement in the “Confidential to Editor” section, and submit your "Accept" recommendation.

Reviewer #1: All comments have been addressed

Reviewer #2: All comments have been addressed

2. Is the manuscript technically sound, and do the data support the conclusions?

Reviewer #1: Yes

Reviewer #2: Yes

3. Has the statistical analysis been performed appropriately and rigorously? 

Reviewer #1: Yes

Reviewer #2: Yes

4. Have the authors made all data underlying the findings in their manuscript fully available?

Reviewer #1: No

Reviewer #2: Yes

5. Is the manuscript presented in an intelligible fashion and written in standard English?

Reviewer #1: Yes

Reviewer #2: Yes

6. Review Comments to the Author

Reviewer #1: the authors have all comments addressed and the paper,

that paper has improved and is suitable for publication

Reviewer #2: (No Response)

7. PLOS authors have the option to publish the peer review history of their article (what does this mean?). If published, this will include your full peer review and any attached files.

Reviewer #1: No

Reviewer #2: No

---

## [Editor Report · Acceptance letter]

15 Jun 2023

PONE-D-22-32871R1 

Increases in income-support payments reduce the demand for charity: A difference-in-difference analysis of charitable-assistance data from Australia over the COVID-19 pandemic 

Dear Dr. Ablaza:

I'm pleased to inform you that your manuscript has been deemed suitable for publication in PLOS ONE. Congratulations! Your manuscript is now with our production department. 

Kind regards, 

on behalf of

Dr. Muhammad Khalid Bashir 

Academic Editor

PLOS ONE